# Multiscale Deep Equilibrium Models

**Shaojie Bai**
Carnegie Mellon University

**Vladlen Koltun**
Intel Labs

**J. Zico Kolter**
Carnegie Mellon University
Bosch Center for AI

## Abstract

We propose a new class of implicit networks, the multiscale deep equilibrium model (MDEQ), suited to large-scale and highly hierarchical pattern recognition domains. An MDEQ directly solves for and backpropagates through the equilibrium points of multiple feature resolutions *simultaneously*, using implicit differentiation to avoid storing intermediate states (and thus requiring only $O(1)$ memory consumption). These simultaneously-learned multi-resolution features allow us to train a *single* model on a diverse set of tasks and loss functions, such as using a single MDEQ to perform both image classification and semantic segmentation. We illustrate the effectiveness of this approach on two large-scale vision tasks: ImageNet classification and semantic segmentation on high-resolution images from the Cityscapes dataset. In both settings, MDEQs are able to match or exceed the performance of recent competitive computer vision models: the first time such performance and scale have been achieved by an implicit deep learning approach. The code and pre-trained models are at `https://github.com/locuslab/mdeq`.

## 1  Introduction

State-of-the-art pattern recognition systems in domains such as computer vision and audio processing are almost universally based on multi-layer hierarchical feature extractors [33, 35, 36]. These models are structured in stages: the input is processed via a number of consecutive blocks, each operating at a different resolution [32, 54, 51, 26]. The architectures explicitly express hierarchical structure, with up- and downsampling layers that transition between consecutive blocks operating at different scales. An important motivation for such designs is the prominent multiscale structure and extremely high signal dimensionalities in these domains. A typical image, for instance, contains millions of pixels, which must be processed coherently by the model.

An alternative approach to differentiable modeling is exemplified by recent progress on *implicit* deep networks, such as Neural ODEs (NODEs) [12] and deep equilibrium models (DEQs) [5]. These constructions replace explicit, deeply stacked layers with analytical conditions that the model must satisfy, and are able to simulate models with "infinite" depth within a constant memory footprint. A notable achievement for implicit modeling is its successful application to large-scale sequences in natural language processing [5].

Is implicit deep learning relevant for general pattern recognition tasks? One clear challenge here is that implicit networks do away with flexible "layers" and "stages". It is therefore not clear whether they can appropriately model multiscale structure, which appears essential to high discriminative power in some domains. This is the challenge that motivates our work. Can implicit models that forego deep sequences of layers and stages attain competitive accuracy in domains characterized by rich multiscale structure, such as computer vision?

To address this challenge, we introduce a new class of implicit networks: the multiscale deep equilibrium model (MDEQ). It is inspired by DEQs, which attained high accuracy in sequence modeling [5]. We expand upon the DEQ construction substantially to introduce simultaneous equilibrium modeling

of multiple signal resolutions. MDEQ solves for equilibria of multiple resolution streams *simultaneously* by directly optimizing for stable representations on *all* feature scales at the same time. Unlike standard explicit deep networks, MDEQ does not process different resolutions in succession, with higher resolutions flowing into lower ones or vice versa. Rather, the different feature scales are maintained side by side in a single "shallow" model that is driven to equilibrium.

This design brings two major advantages. First, like the basic DEQ, our model does not require backpropagation through an explicit stack of layers and has an $O(1)$ memory footprint during training. This is especially important as pattern recognition systems are memory-intensive. Second, MDEQ rectifies one of the drawbacks of DEQ by exposing multiple feature scales at equilibrium, thereby providing natural interfaces for auxiliary losses and for compound training procedures such as pretraining (e.g., on ImageNet) and fine-tuning (e.g., on segmentation or detection tasks). Multiscale modeling enables a *single* MDEQ to simultaneously train for multiple losses defined on potentially very different scales, whose equilibrium features can serve as "heads" for a variety of tasks.

We demonstrate the effectiveness of MDEQ via extensive experiments on large-scale image classification and semantic segmentation datasets. Remarkably, this shallow implicit model attains comparable accuracy levels to state-of-the-art deeply-stacked explicit ones. On ImageNet classification, MDEQs outperform baseline ResNets (e.g., ResNet-101) with similar parameter counts, reaching 77.5% top-1 accuracy. On Cityscapes semantic segmentation (dense labeling of 2-megapixel images), identical MDEQs to the ones used for ImageNet experiments match the performance of recent explicit models while consuming much less memory. Our largest MDEQ surpasses 80% mIoU on the Cityscapes validation set, outperforming strong convolutional networks and coming tantalizingly close to the state of the art. This is by far the largest-scale application of implicit deep learning to date and a remarkable result for a class of models that until recently were applied largely to "toy" domains.

## 2   Background

**Implicit Deep Learning.**   Virtually all modern deep learning approaches use *explicit* models, which provide explicit *computation graphs* for forward propagation. Backward passes proceed in reverse order through the same graph. This approach is the core of popular deep learning frameworks [1] and is associated with the very concept of "architecture". In contrast, *implicit* models do not have prescribed computation graphs. They instead posit a specific criterion that the model must satisfy (e.g., the endpoint of an ODE flow, or the root of an equation). Importantly, the algorithm that drives the model to fulfill this criterion is not prescribed. Therefore, implicit models can leverage black-box solvers in their forward passes and enjoy analytical backward passes that are independent of the forward pass trajectories.

Implicit modeling of hidden states has been explored by the deep learning community for decades. Pineda [43] and Almeida [2] studied implicit differentiation techniques for training recurrent dynamics, also known as recurrent back-propagation (RBP) [37]. Implicit approaches to network design have recently attracted renewed interest [20, 24]. For example, Neural ODEs (NODEs) [12, 18] model a recursive residual block using implicit ODE solvers, equivalent to a continuous ResNet taking infinitesimal steps. Deep equilibrium models (DEQs) [5] solve for the fixed point of a sequence model with black-box root-finding methods, equivalent to finding the limit state of an infinite-layer network. Other instantiations of implicit modeling include optimization layers [17, 3], differentiable physics engines [14, 45], logical structure learning [58], and continuous generative models [25].

Our work takes the deep equilibrium approach [5] into signal domains characterized by rich multiscale structure. We develop the first one-layer implicit deep model that is able to scale to realistic visual tasks (e.g., megapixel-level images), and achieve competitive results in these regimes. In comparison, ODE-based models have so far only been applied to relatively low-dimensional signals due to numerical instability. For example, Chen et al. [12] downsampled $28 \times 28$ MNIST images to $7 \times 7$ before feeding them to Neural ODEs. More broadly, our work can be seen as a new perspective on implicit models, wherein the models define and optimize simultaneous criteria over multiple data streams that can have different dimensionalities. While DEQs and NODEs have so far been defined on a single stream of features, a single MDEQ can jointly optimize features for different tasks, such as image segmentation and classification.

**Multiscale Modeling in Computer Vision.**   Computer vision is a canonical application domain for hierarchical multiscale modeling. The field has come to be dominated by deep convolutional

networks [33, 32]. Computer vision problems can be viewed in terms of the granularity of the desired output: from low-resolution, such as a label for a whole image [16], to high-resolution output that assigns a label to each pixel, as in semantic segmentation [49, 11, 61, 64]. State-of-the-art models for these problems are explicitly structured into sequential stages of processing that operate at different resolutions [32, 54, 51, 26]. For example, a ResNet [26] typically consists of 4-6 sequential stages, each operating at half the resolution of the preceding one. A dilated ResNet [62] uses a different schedule for the progression of resolutions. A DenseNet [27] uses different connectivity patterns to carry information between layers, but shares the overarching structure: a sequence of stages. Other designs progressively decrease feature resolution and then increase it step by step [46]. Downsampling and upsampling can also be repeated, again in an explicitly choreographed sequence [42, 53].

Multiscale modeling has been a central motif in computer vision. The Laplacian pyramid is an influential early example of multiscale modeling [7]. Multiscale processing has been integrated with convolutional networks for scene parsing by Farabet et al. [21] and has been explicitly addressed in many subsequent architectures [49, 11, 61, 8, 38, 64, 28, 10, 57].

Our work brings multiscale modeling to *implicit* deep networks. MDEQ has in essence only *one* stage, in which the different resolutions coexist side by side. The input is injected at the highest resolution and then propagated implicitly to the other scales, which are optimized simultaneously by a (black-box) solver that drives them to satisfy a joint equilibrium condition. Just like DEQs, an MDEQ is able to represent an "infinitely" deep network with only a constant memory cost.

## 3 Multiscale Deep Equilibrium Models

We begin by briefly summarizing the basic DEQ construction and some major challenges that arise when extending it to computer vision.

### 3.1 Deep Equilibrium (DEQ): Generic Formulation

One of the core ideas that motivated the DEQ approach was weight-tying: the same set of parameters can be shared across the layers of a deep network. Formally, Bai et al. [5] formulated an $L$-layer weight-tied transformation with parameter $\theta$ on hidden state $\mathbf{z}$ as

$$\mathbf{z}^{[i+1]} = f_\theta(\mathbf{z}^{[i]}; \mathbf{x}), \quad i = 0, \ldots, L-1 \tag{1}$$

where the input representation $\mathbf{x}$ was injected into each layer. When sufficient stability conditions were ensured, stacking such layers infinitely (i.e., $L \to \infty$) was shown to essentially perform fixed-point iterations and thus tend to an equilibrium $\mathbf{z}^\star = f_\theta(\mathbf{z}^\star; \mathbf{x})$. Intuitively, as we iterate the transformation $f_\theta$, the hidden representation tends to converge to a stable state, $\mathbf{z}^\star$. Such construction has a number of appealing properties. First, we can directly solve for the fixed point, which can be done faster than explicitly iterating through the layers. We formulate this as a root-finding problem:

$$g_\theta(\mathbf{z}; \mathbf{x}) \coloneqq f_\theta(\mathbf{z}; \mathbf{x}) - \mathbf{z} \implies \mathbf{z}^\star = \mathsf{Rootfind}(g_\theta; \mathbf{x}) \tag{2}$$

For example, one can leverage Newton or quasi-Newton methods to achieve quadratic or superlinear convergence to the root. Second, one can directly backpropagate through the equilibrium state using the Jacobian of $g_\theta$ at $\mathbf{z}^\star$, without tracing through the forward root-finding process. Formally, given a loss $\ell = \mathcal{L}(\mathbf{z}^\star, \mathbf{y})$ (where $\mathbf{y}$ is the target), the gradients can be written as

$$\frac{\partial \ell}{\partial \theta} = \frac{\partial \ell}{\partial \mathbf{z}^\star} \left(-J_{g_\theta}^{-1}|_{\mathbf{z}^\star}\right) \frac{\partial f_\theta(\mathbf{z}^\star; \mathbf{x})}{\partial \theta} \qquad \frac{\partial \ell}{\partial \mathbf{x}} = \frac{\partial \ell}{\partial \mathbf{z}^\star} \left(-J_{g_\theta}^{-1}|_{\mathbf{z}^\star}\right) \frac{\partial f_\theta(\mathbf{z}^\star; \mathbf{x})}{\partial \mathbf{x}}. \tag{3}$$

See Bai et al. [5] for the proof, which is based on the implicit function theorem [30]. This means that the forward pass of a DEQ can rely on any black-box root solver, while the backward pass is based independently on differentiating through only one layer (or block) at the equilibrium (i.e., $\frac{\partial f_\theta(\mathbf{z}^\star; \mathbf{x})}{\partial (\cdot)}$). The memory consumption of the entire training process is equivalent to that of just one block rather than $L \to \infty$ blocks. Since the Jacobian of $g_\theta$ can be expensive to compute, DEQs solve for a linear equation involving a vector-Jacobian product, which is a lot cheaper:

$$\mathbf{x}(J_{g_\theta}|_{\mathbf{z}^\star}) + \frac{\partial \ell}{\partial \mathbf{z}^\star} = \mathbf{0}. \tag{4}$$

The DEQ model therefore solves for the network output at its *infinite depth*, with each step of the model now implicitly defined to reach an analytical objective (the equilibrium).

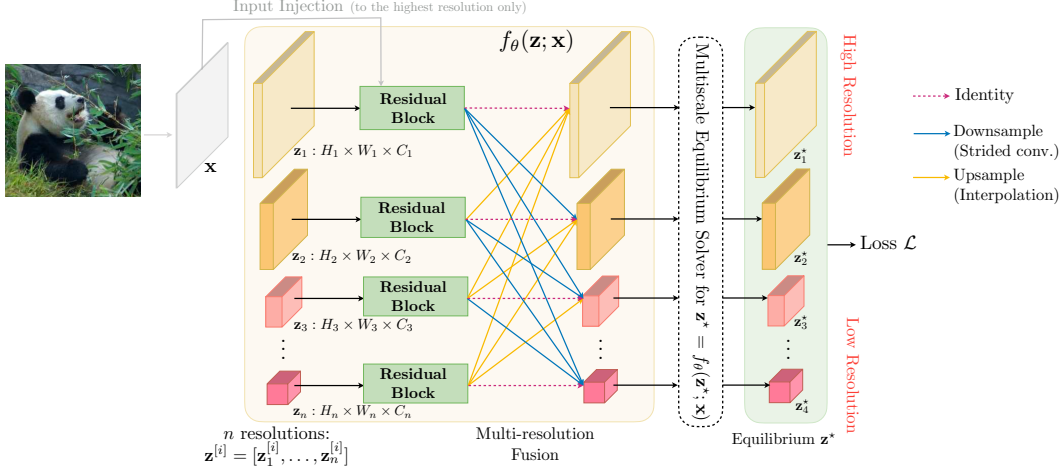

Figure 1: The structure of a multiscale deep equilibrium model (MDEQ). *All components* of the model are shown in this figure. MDEQ consists of a transformation $f_\theta$ that is driven to equilibrium. Features at different scales coexist side by side and are driven to equilibrium simultaneously.

**Challenges.** The construction of Bai et al. [5], which we have just summarized, was primarily aimed at processing *sequences*. As we transition from sequences to high-resolution images, we note important differences between these domains. First, unlike typical autoregressive sequence learning problems (e.g., language modeling), where input and output have identical length and dimensionality, general pattern recognition systems (such as those in vision) entail multi-stage modeling via a combination of up- and downsampling in the architecture. The basic DEQ construction does not exhibit such structure. Second, the output of a computer vision task such as image classification (a label) or object localization (a region) may have very different dimensionality from the input (a full image): again a feature that the basic DEQ does not support. Third, state-of-the-art models for tasks such as semantic segmentation are commonly based on "backbones" that are pretrained for image classification, even though the tasks are structurally different and their outputs have very different dimensionalities (e.g., one label for the whole image versus a label for each pixel). It's not clear how a DEQ construction can support such transfer. Fourth, whereas past work on DEQs could leverage state-of-the-art weight-tied architectures for sequence modeling as the basis for the transformation $f_\theta$ [4, 15], no such counterparts exist in state-of-the-art computer vision modeling.

## 3.2 The MDEQ Model

**Notation.** Figure 1 illustrates the *entire* structure of MDEQ. As before, $f_\theta$ denotes the transformation that is (implicitly) iterated to a fixed point, $\mathbf{x}$ is the (precomputed) input representation provided to $f_\theta$, and $\mathbf{z}$ is the model's internal state. We omit the batch dimension for clarity.

**Transformation $f_\theta$.** The central part of MDEQ is the transformation $f_\theta$ that is driven to equilibrium. We use a simple design in which features at each resolution are first taken through a residual block. The blocks are shallow and are identical in structure. At resolution $i$, the residual block receives the internal state $\mathbf{z}_i$ and outputs a transformed feature tensor $\mathbf{z}_i^+$ at the same resolution. Notably, the highest resolution stream (i.e., $i = 1$) also receives an input injection $\mathbf{x}$ that is precomputed directly from the source image and injected to the highest-resolution residual block. (See Eq. (5) and the discussion below.)

The internal structure of the residual block is shown in Figure 2. We largely adopt the design of He et al. [26], but use group normalization [59] rather than batch normalization [29], for stability reasons that are discussed in Section 3.3. The residual block at resolution $i$ can be formally expressed as

$$\begin{aligned}
\tilde{\mathbf{z}}_i &= \text{GroupNorm}\big(\text{Conv2d}(\mathbf{z}_i)\big) \\
\hat{\mathbf{z}}_i &= \text{GroupNorm}\big(\text{Conv2d}(\text{ReLU}(\tilde{\mathbf{z}}_i)) + \mathbb{1}_{\{i=1\}} \cdot \mathbf{x}\big) \\
\mathbf{z}_i^+ &= \text{GroupNorm}\big(\text{ReLU}(\hat{\mathbf{z}}_i + \mathbf{z}_i)\big)
\end{aligned} \qquad (5)$$

Following these blocks, the second part of $f_\theta$ is a multi-resolution fusion step that mixes the feature maps across different scales (see Figure 1). The transformed features $\mathbf{z}_i^+$ undergo either upsampling

or downsampling from the current scale $i$ to each other scale $j \neq i$. In our construction, downsampling is performed by $j - i$ consecutive 2-strided $3 \times 3$ Conv2d, whereas upsampling is performed by direct bilinear interpolation. The final output at scale $j$ is formed by summing over the transformed feature maps provided from all incoming scales $i$ (along with $\mathbf{z}_j^+$); i.e., the output feature tensor at each scale is a mixture of transformed features from all scales. This forces the features at all scales to be consistent and drives the whole system to a coordinated equilibrium that harmonizes the representations across scales.

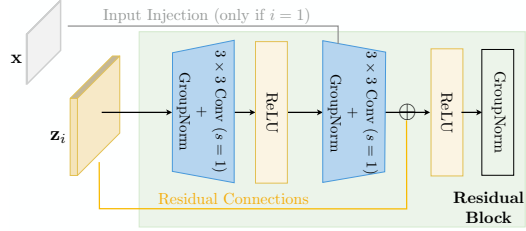

Figure 2: The residual block used in MDEQ. An MDEQ contains only *one* such layer.

**Input Representation.** The raw input first goes through a transformation (e.g., a linear layer that aligns the feature channels) to form $\mathbf{x}$, which will be provided to $f_\theta$. The existence of such input injection is vital to implicit models as it (along with $\theta$) correlates the flow of the dynamical system with the input. However, unlike multiscale input representations used by some explicit vision architectures [21, 13], we only inject $\mathbf{x}$ to the highest-resolution feature stream (see Eq. (5)). The input is provided to MDEQ at a single (full) resolution. The lower resolutions hence start with no knowledge at all about the input; this information will only *implicitly* propagate through them as all scales are gradually driven to coordinated equilibria $\mathbf{z}^\star$ by the (black-box) solver.

**(Limited-memory) Multiscale Equilibrium Solver.** In the DEQ, the internal state is a single tensor $\mathbf{z}$ [5]. The MDEQ state, however, is a *collection* of tensors at $n$ resolutions: $\mathbf{z} = [\mathbf{z}_1, \ldots, \mathbf{z}_n]$. Note that this is not a concatenation, as the different $\mathbf{z}_i$ have different dimensionalities, feature resolutions, and semantics.

With this in mind, our equilibrium solver leverages Broyden's method. We initialize the internal states by setting $\mathbf{z}_i^{[0]} = \mathbf{0}$ for all scales $i$. $\mathbf{z} = [\mathbf{z}_1, \ldots, \mathbf{z}_n]$ is maintained as a collection of $n$ tensors whose respective equilibrium states (i.e., roots) are solved for and backpropagated through simultaneously (with each resolution inducing its own loss).

The original Broyden solver was not efficient enough when applied to computer vision datasets, which have very high dimensionality. For example, in the Cityscapes segmentation task (see Section 4), the Jacobian of a 4-resolution MDEQ at $\mathbf{z}^\star$ is well over 2,000 times larger than its single-scale counterpart in word-level language modeling [5]. Note that even with low-rank approximations of the Jacobian in quasi-Newton methods, the high dimensionality of images can make storing these updates extremely expensive. To address this, we improve the memory efficiency of the forward and backward passes by optimizing Broyden's method. We implemented a new solver that is inspired by Limited-memory BFGS (L-BFGS) [39], where we only keep the latest $m$ low-rank updates at any step and discard the earlier ones (see Appendix B.1).

**Pretraining and Auxiliary Losses.** Figure 3 provides a comparison of MDEQ with single-stream implicit models such as the DEQ, and with explicit deep networks in computer vision. These different models expose different "interfaces" that can be used to define losses for different tasks. Prior implicit models such as neural ODEs and DEQs typically assume that a loss is defined on a single stream of implicit hidden states, which has a uniform input and output shape (Figure 3b). It is therefore not clear how such a model can be flexibly transferred across structurally different tasks (e.g., pretraining on image classification and fine-tuning on semantic segmentation). Furthermore, there is no natural way to define auxiliary losses [34], because there are no "layers" and the forward and backward computation trajectories are decoupled.

In comparison, MDEQ exposes convenient "interfaces" to its states at multiple resolutions. One resolution (the highest) can be the same as the resolution of the input, and can be used to define losses for dense prediction tasks such as semantic segmentation. Another resolution (the lowest) can be a vector in which the spatial dimensions are collapsed, and can be used to define losses for image-level labeling tasks such as image classification. This suggests clean protocols for training the same model for different tasks, either jointly (e.g., multi-task learning in which structurally different supervision flows through multiple heads) or in sequence (e.g., pretraining for image classification through one head and fine-tuning for semantic segmentation through another).

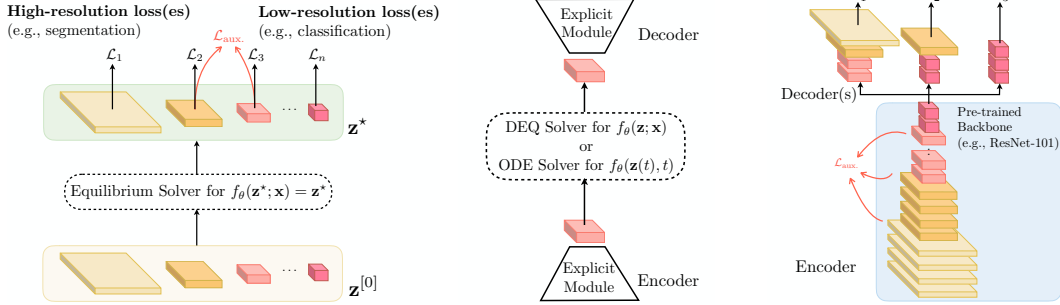

(a) MDEQ exposes multiple interfaces at equilibrium

(b) Single-stream implicit models (e.g., DEQs and NODEs)

(c) Explicit deep models in vision

Figure 3: A visual comparison of MDEQ with prior implicit models and with standard explicit models in computer vision. Equilibrium states at multiple resolutions enable MDEQ to incorporate supervision in different forms.

## 3.3 Integrating Common DL Techniques with MDEQs

MDEQ simulates an "infinitely" deep network by implicitly modeling one layer. Such implicitness calls for care when adapting common deep learning practices. We provide an exploration of such adaptations and their impact on the training dynamics of MDEQ. We believe these observations will also be valuable for future research on implicit models.

**Normalization.** Layer normalization of hidden activations in $f_\theta$ played an important role in constraining the output and stabilizing DEQs on sequences [5]. A natural counterpart in vision is batch normalization (BN) [29]. However, BN is not directly suitable for implicit models, since it estimates population statistics based on layers, which are implicit in our setting, and the Jacobian matrix of the transformation $f_\theta$ will scale badly to make the fixed point significantly harder to solve for. We therefore use group normalization (GN) [59], which groups the input channels and performs normalization within each group. GN is independent of batch size and offers more natural support for transfer learning (e.g., pretraining and fine-tuning on structurally different tasks). Unlike in DEQs, we keep the learnable affine parameters of GN.

**Dropout.** The conventional spatial dropout used by explicit vision models applies a random mask to given layers in the network [52]. A new mask is generated whenever dropout is invoked. Such layer-based stochasticity can significantly hurt the stability of convergence to the equilibrium. In fact, as two adjacent calls to $f_\theta$ most probably will have different Bernoulli dropout masks, it is almost impossible to reach a fixed point where $f_\theta(\mathbf{z}^\star; \mathbf{x}) = \mathbf{z}^\star$. We therefore adopt variational dropout [22] and apply the exact same mask at all invocations of $f_\theta$ in a given training iteration. The mask is reset at each training iteration.

**Nonlinearities.** The multiscale features are initialized to $\mathbf{z}_i^{[0]} = \mathbf{0}$ for all resolutions $i$. However, we found that this could induce certain instabilities when training MDEQ (especially in the starting phase of it), most likely due to the drastic change of slope of the ReLU non-linearity at the origin, where the derivative is undefined [23]. To combat this, we replace the last ReLU in both the residual block and the multiscale fusion by a softplus [23] in the initial phase of training. These are later switched back to ReLU. The softplus provides a smooth approximation to the ReLU, but has slope $1 - \frac{1}{1+\exp(\beta \mathbf{z})} \to \frac{1}{2}$ around $\mathbf{z} = 0$ (where $\beta$ controls the curvature).

**Convolution and Convergence to Equilibrium.** Whereas the original DEQ model focused primarily on self-attention transformations [56], where all hidden units communicate globally, MDEQ models face additional challenges due to the nature of typical vision models. Specifically, our MDEQ models employ convolutions with small receptive fields (e.g., the two $3 \times 3$ convolutional filters in $f_\theta$'s residual block) on potentially very large images: for instance, we eventually evaluate our semantic segmentation model on megapixel-scale images. In consequence, we typically need a higher number of root-finding iterations to converge to an exact equilibrium. While this does pose a challenge, we find that using the aforementioned strategies of 1) multiscale simultaneous up- and downsampling and 2) quasi-Newton root-finding, drives the model close to equilibrium within a reasonable number of iterations. We further analyze convergence behavior in Appendix B.

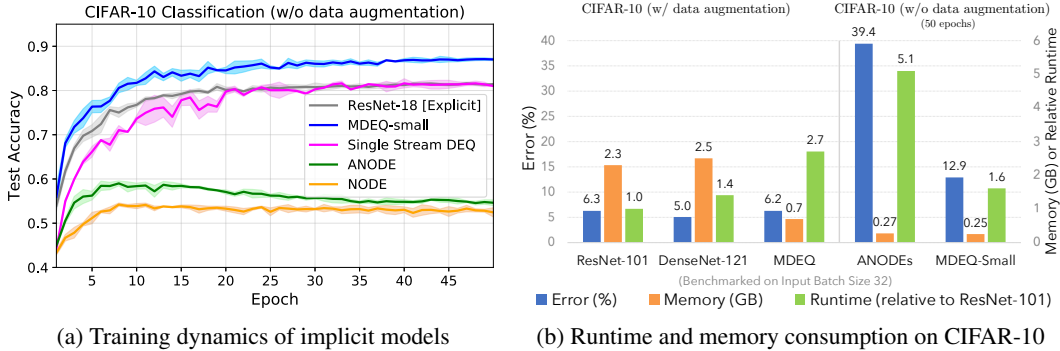

(a) Training dynamics of implicit models        (b) Runtime and memory consumption on CIFAR-10

Figure 4: Left: test accuracy as a function of training epochs. Right: MDEQ-Small and ANODEs correspond to the settings and results reported in Table 1. For all metrics, lower is better.

# 4   Experiments

In this section, we investigate the empirical performance of MDEQs from two aspects. First, as prior implicit approaches such as NODEs have mostly evaluated on smaller-scale benchmarks such as MNIST [33] and CIFAR-10 ($32 \times 32$ images) [31], we compare MDEQs with these baselines on the same benchmarks. We evaluate both training-time stability and inference-time performance. Second, we evaluate MDEQs on large-scale computer vision tasks: ImageNet classification [16] and semantic segmentation on the Cityscapes dataset [13]. These tasks have

Table 1: Evaluation on CIFAR-10. Standard deviations are calculated on 5 runs.

| | Model Size | Accuracy |
|---|---|---|
| CIFAR-10 (*without* data augmentation) | | |
| Neural ODEs [18] | 172K | 53.7% $\pm$ 0.2% |
| Aug. Neural ODEs [18] | 172K | 60.6% $\pm$ 0.4% |
| Single-stream DEQ [5] | 170K | 82.2% $\pm$ 0.3% |
| ResNet-18 [26] [Explicit] | 170K | 81.6% $\pm$ 0.3% |
| **MDEQ-small (ours)** | **170K** | **87.1%** $\pm$ 0.4% |
| CIFAR-10 (*with* data augmentation) | | |
| ResNet-18 [26] [Explicit] | 10M | **92.9%** $\pm$ 0.2% |
| **MDEQ (ours)** | **10M** | **93.8%** $\pm$ 0.3% |

extremely high-dimensional inputs (e.g., $2048 \times 1024$ images for Cityscapes) and are dominated by explicit models. We provide more detailed descriptions of the tasks, hyperparameters, and training settings in Appendix A.

Our focus is on the behavior of MDEQs and their competitiveness with prior implicit or explicit models. We are not aiming to set a new state of the art on ImageNet classification or Cityscapes segmentation, as this typically involves substantial additional investment [60]. However, we do note that even with the implicit modeling of layer $f_\theta$, the *mini explicit structure* within the design of $f_\theta$ (e.g., the residual block) is still very helpful empirically in improving the equilibrium representations.

All experiments with MDEQs use the limited-memory version of Broyden's method in both forward and backward passes, and the root solvers are stopped whenever 1) the objective value reaches some predetermined threshold $\varepsilon$ or 2) the solver's iteration count reaches a limit $T$. On large-scale vision benchmarks (ImageNet and Cityscapes), we downsample the input twice with 2-strided convolutions before feeding it into MDEQs, following the common practice in explicit models [64, 57]. We use the cosine learning rate schedule for all tasks [41].

## 4.1   Comparing with Prior Implicit Models on CIFAR-10

Following the setting of Dupont et al. [18], we run the experiments on CIFAR-10 classification (without data augmentation) for 50 epochs and compare models with approximately the same number of parameters. However, unlike the ODE-based approaches, we do not perform downsamplings on the raw images before passing the inputs to the MDEQ solver (so the highest-resolution stream stays at $32 \times 32$). When training the MDEQ model, *all* resolutions are used for the final prediction: higher-resolution streams go through additional downsampling layers and are added to the lowest-resolution output to make a prediction (i.e., a form of auxiliary loss).

The results of MDEQ models on CIFAR-10 image classification are shown in Table 1. Compared to NODEs [12] and Augmented NODEs [18], a small MDEQ with a similar parameter count improves accuracy by more than 20 percentage points: an error reduction by *more than a factor of 2*. MDEQ also improves over the single-stream DEQ (applied at the highest resolution). The training dynamics of the different models are visualized in Figure 4a. Finally, a larger MDEQ matches and even

Table 2: Evaluation on ImageNet classification with top-1 and top-5 accuracies reported. MDEQs were trained for 100 epochs.

| | Model Size | top1 Acc. | top5 Acc. |
|---|---|---|---|
| AlexNet [32] | 238M | 57.0% | 80.3% |
| ResNet-18 [26] | 13M | 70.2% | 89.9% |
| ResNet-34 [26] | 21M | 74.8% | 91.1% |
| Inception-V2 [29] | 12M | 74.8% | 92.2% |
| ResNet-50 [26] | 26M | 75.1% | 92.5% |
| HRNet-W18-C [57] | 21M | 76.8% | 93.4% |
| Single-stream DEQ + global pool [5] | 18M | 72.9% | 91.0% |
| **MDEQ-small (ours) [Implicit]** | 18M | 75.5% | 92.7% |
| ResNet-101 [26] | 52M | 77.1% | 93.5% |
| W-ResNet-50 [63] | 69M | 78.1% | 93.9% |
| DenseNet-264 [27] | 74M | 79.7% | 94.8% |
| **MDEQ-large (ours) [Implicit]** | 63M | 77.5% | 93.6% |
| Unrolled 5-layer *MDEQ-large* | 63M | 75.9% | 93.0% |
| **MDEQ-XL (ours) [Implicit]** | 81M | 79.2% | 94.5% |

Table 3: Evaluation on Cityscapes `val` semantic segmentation. "*" marks the current SOTA. Higher mIoU (mean Intersection over Union) is better.

| | Backbone | Model Size | mIoU |
|---|---|---|---|
| ResNet-18-A [40] | ResNet-18 | 3.8M | 55.4 |
| ResNet-18-B [40] | ResNet-18 | 15.24M | 69.1 |
| MobileNetV2Plus [48] | MobileNetV2 | 8.3M | 74.5 |
| GSCNN [55] | ResNet-50 | - | 73.0 |
| HRNetV2-W18-Small-v2* [57] | HRNet | 4.0M | 76.0 |
| **MDEQ-small (ours) [Implicit]** | MDEQ | 7.8M | 75.1 |
| U-Net++ [66] | ResNet-101 | 59.5M | 75.5 |
| Dilated-ResNet [62] | D-ResNet-101 | 52.1M | 75.7 |
| PSPNet [64] | D-ResNet-101 | 65.9M | 78.4 |
| DeepLabv3 [9] | D-ResNet-101 | 58.0M | 78.5 |
| PSANet [65] | ResNet-101 | - | 78.6 |
| HRNetV2-W48* [57] | HRNet | 65.9M | 81.1 |
| **MDEQ-large (ours) [Implicit]** | MDEQ | 53.0M | 77.8 |
| **MDEQ-XL (ours) [Implicit]** | MDEQ | 70.9M | 80.3 |

exceeds the accuracy of a ResNet-18 with the same capacity: the first time such performance has been demonstrated by an implicit model.

## 4.2 ImageNet Classification

We now test the ability of MDEQ to scale to a much larger dataset with higher-resolution images: ImageNet [16]. As with CIFAR-10 classification, we add a shallow classification layer after the MDEQ module to fuse the equilibrium outputs from different scales, and train on a combined loss.

We benchmark both a small MDEQ model and a large MDEQ to provide appropriate comparisons with a number of reference models, such as ResNet-18, -34, -50, and -101 [26]. Note that MDEQ has only *one layer* of residual blocks followed by multi-resolution fusion. Therefore, to match the capacity of standard explicit models, we need to increase the feature dimensionality within MDEQ. This is accomplished mainly by adjusting the width of the convolutional filter within the residual block (see Figure 2).

Table 2 shows the accuracy of two MDEQs (of different sizes) in comparison to well-known reference models in computer vision. MDEQs are remarkably competitive with strong explicit models. For example, a small MDEQ with 18M parameters outperforms ResNet-18 (13M parameters), ResNet-34 (21M parameters), and even ResNet-50 (26M parameters). A larger MDEQ (64M parameters) reaches the same level of performance as ResNet-101 (52M parameters). This is far beyond the scale and accuracy levels of prior applications of implicit modeling.

## 4.3 Cityscapes Semantic Segmentation

After training on ImageNet, we train *the same MDEQs* for semantic segmentation on the Cityscapes dataset [13]. When transferring the models from ImageNet to Cityscapes, we directly use the highest-resolution equilibrium output $z_1^\star$ to train on the highest-resolution loss. Thus *MDEQ is its own "backbone"*. We train on the Cityscapes `train` set and evaluate on the `val` set. Following the evaluation protocol of Zhao et al. [65] and Wang et al. [57], we test on a single scale with no flipping.

MDEQs attain remarkably high levels of accuracy. They come close to the current state of the art, and match or outperform well-known and carefully architected explicit models that were released in the past two years. A small MDEQ (7.8M parameters) achieves a mean IoU of 75.1. This improves upon a MobileNetV2Plus [48] of the same size and is close to the SOTA for models on this scale. A large MDEQ (53.5M parameters) reaches 77.8 mIoU, which is within 1 percentage point of highly regarded recent semantic segmentation models such as DeepLabv3 [9] and PSPNet [64], whereas a larger version (70.9M parameters) surpasses them. It is surprising that such levels of accuracy can be achieved by a "shallow" implicit model, based on principles that have not been applied to this domain before. Examples of semantic segmentation results are shown in Appendix C.

## 4.4 Runtime and Memory Consumption

We provide a runtime and memory analysis of MDEQs using CIFAR-10 data, with input batch size 32. Since prior implicit models such as ANODEs [18] are relatively small, we provide results for both MDEQ and MDEQ-small for a fair comparison. All computation speeds are benchmarked relative

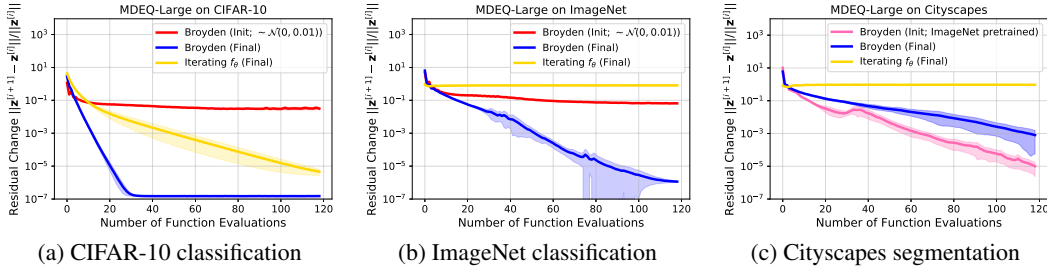

| (a) CIFAR-10 classification | (b) ImageNet classification | (c) Cityscapes segmentation |

Figure 5: Plots of MDEQ's convergence to equilibrium (measured by $\frac{\|\mathbf{z}^{[i+1]}-\mathbf{z}^{[i]}\|}{\|\mathbf{z}^{[i]}\|}$) as a function of the number of times we evaluate $f_\theta$. As input image resolution grows (from CIFAR-10 to Cityscapes), MDEQ takes more steps to converge with (L-)Broyden's method. Standard deviation is calculated on 5 randomly selected batches from each dataset.

to the ResNet-101 model (about 150ms per batch) on a single RTX 2080 Ti GPU. The results are summarized in Figure 4b.

MDEQ saves more than 60% of the GPU memory at training time compared to explicit models such as ResNets and DenseNets, while maintaining competitive accuracy. Training a large MDEQ on ImageNet consumes about 6GB of memory, which is mostly used by Broyden's method. This low memory footprint is a direct result of the analytical backward pass. Meanwhile, MDEQs are generally slower than explicit networks. We observe a $2.7\times$ slowdown for MDEQ compared to ResNet-101, a tendency similar to that observed in the sequence domain [5]. A major factor contributing to the slowdown is that MDEQs maintain features at all resolutions throughout, whereas explicit models such as ResNets gradually downsample their activations and thus reduce computation (e.g., 70% of ResNet-101 layers operate on features that are downsampled by $8 \times 8$ or more). However, when compared to ANODEs with 172K parameters, an MDEQ of similar size is $3\times$ *faster while achieving a $3\times$ error reduction*. Additional discussion of runtime and convergence is provided in Appendix B.2.

### 4.5 Equilibrium Convergence on High-resolution Inputs

As we scale MDEQ to higher-resolution inputs, the equilibrium solving process becomes more challenging. This is illustrated in Figure 5, where we show the equilibrium convergence of MDEQ on CIFAR-10 (low-resolution), ImageNet (medium-resolution) and Cityscapes (high-resolution) images by measuring the change of residual with respect to the number of function evaluations. We empirically find that (limited-memory) Broyden's method and multiscale fusion both help stabilize the convergence on high-resolution data. For example, in all three cases, Broyden's method (blue lines in Figure 5) converges to the fixed point in a more stable and efficient manner than simply iterating $f_\theta$ (yellow lines). Further analysis of the multiscale convergence behavior is provided in Appendix B.2.

## 5 Conclusion

We introduced multiscale deep equilibrium models (MDEQs): a new class of implicit architectures for domains characterized by high dimensionality and multiscale structure. Unlike prior implicit models, such as DEQs and Neural ODEs, an MDEQ solves for and backpropagates through synchronized equilibria of multiple feature representations at different resolutions. We show that a single MDEQ can be used for different tasks, such as image classification and semantic segmentation. Our experiments demonstrate for the first time that "shallow" implicit models can scale to practical computer vision tasks and achieve competitive performance that matches explicit architectures characterized by sequential processing through deeply stacked layers.

The remarkable performance of implicit models in this work brings up core questions in machine learning. Are complex stage-wise hierarchical architectures, which have dominated deep learning to date, necessary? MDEQ exemplifies a different approach to differentiable modeling. The most significant message of our work is that this approach may be much more relevant in practice than previously appeared. We hope that this will contribute to the development of implicit deep learning and will further broaden the agenda in differentiable modeling.

## Broader Impacts

Computer vision techniques themselves, which are the primary application focus on this paper, have numerous applications of both positive and negative societal benefits. They can enable potentially live-saving advances in e.g., assisted driving, medical diagnoses, etc, but also have inherent limitations and biases that could lead to problematic applications in these areas (e.g., if performance of a vision model notably differs when given input images of people of different races or genders); and this says nothing of more genuinely problematic enabled applications, such as facial recognition for surveillance applications.

The question of more relevance to this paper, however, is whether there any societal-level consequences that are unique to *this particular* algorithmic approach, i.e., the use of implicit versus explicit models in computer vision domains. This point is genuinely less clear to us. It is possible that the relative memory-efficiency of implicit vision models would make them e.g., more amenable to edge devices, which in turn contribution raises the potential for both beneficial and harmful use cases. However, this is a large leap from current methods, where the improved memory efficiency comes at a cost of increased compute time (and thus could arguably be *less* efficient in their current form on edge devices). Thus, we believe the specific impacts of implicit models are still unclear at this point, and should largely be re-evaluated as the models become more standard or more widely adopted.

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
