[Supplementary Material]

# A  Task Descriptions and Training Settings

We provide a detailed description of all tasks and some additional details on the training of MDEQ.

**Image Classification on CIFAR-10.**  CIFAR-10 is a well-known computer vision dataset that consists of 60,000 color images, each of size $32 \times 32$ [31]. There are 10 object classes and 6,000 images per class. The entire dataset is divided into training (50K images) and testing (10K) sets.

We use two different training settings for evaluating the MDEQ model on CIFAR-10. Following Dupont et al. [18], we compare MDEQ-small with other implicit models on CIFAR-10 images *without data augmentation* (i.e., the original, raw images), using approximately 170K learnable parameters in the model. In the second setting, we apply data augmentation to the input images (i.e., random cropping, horizontal flipping, etc.), a setting that most competitive vision baselines (e.g., ResNets) use by default.

**Image Classification on ImageNet.**  The dataset we use contains 1.2 million labeled training images from ImageNet [32] distributed over 1,000 classes, and a test set of 150,000 images. The original ImageNet consists of variable-resolution images, and we follow the standard setting [26] to use the $224 \times 224$ crops as inputs to the model.

ImageNet is frequently used for pretraining general-purpose image feature extractors that are used on downstream tasks [26, 63, 62, 57]. We train a small and large MDEQ model, which will act as their own "backbone" when later fine-tuned on the Cityscapes segmentation task. We train MDEQ on ImageNet for 100 epochs. Following the practice of Bai et al. [5] with DEQ models for sequences, we start the training (the first few epochs) of MDEQ with a shallow (5-layer) weight-tied stacking of $f_\theta$ to warm up the weights, and then switch to the implicit equilibrium (root) solver for the rest of the training epochs.

**Semantic Segmentation on Cityscapes.**  Cityscapes is a large-scale urban scene understanding dataset containing high-quality, pixel-level annotated street scene images from 50 cities [13]. The dataset consists of 5,000 images, which are divided into 2,975 (`train`), 500 (`val`) and 1,525 (`test`) sets. Each pixel is classified in a 19-way fashion for evaluation.

We follow the training protocol of prior works [64, 57] to train the MDEQ models on the Cityscapes `train`, and perform random cropping (to $1024 \times 512$) and random horizontal flipping on the training inputs. The models are evaluated on the Cityscapes `val` (single scale and no random flipping) with the original resolution $2048 \times 1024$. We use the identical MDEQ model(s) as used in ImageNet training, but now predict with the high-resolution head.

**Hyperparameters.**  We provide the hyperparameters of the models we used in each of these tasks in Table 4. Note that we use a *single* model for both ImageNet classification and Cityscapes segmentation, so the models share the same configuration (highlighted in red in Table 4 for clarity). For all tasks, the MDEQ features in resolution $i = 1, \ldots, n$ take the shape $\left(\frac{H}{2^{i-1}}, \frac{W}{2^{i-1}}\right)_{i=1,\ldots,n}$, where $H, W$ are the dimensions of the original input. In other words, each resolution uses half the feature size of its next higher resolution stream. We apply weight normalization [47] to all of the learnable weights in $f_\theta$.

**Hardware.**  For both ImageNet and Cityscapes experiments, MDEQ-Large models were trained on 4 RTX-2080 Ti GPUs, while MDEQ-XL models were trained on 8 Quadro RTX 8000 GPUs. The CIFAR-10 classification models were trained on 1 GPU (including the baselines).

**Initialization of MDEQ Models.**  For CIFAR-10 and ImageNet, we initialize the parameters of $f_\theta$ randomly from $\mathcal{N}(0, 0.01)$ (Cityscapes MDEQs use pretrained ImageNet MDEQs). Generally, we observe that the final performance of MDEQ is not sensitive to the choice of initialization distribution. However, such random initialization could occasionally induce instabilities in the starting phase of the training (see red lines in Figure 6). We solve this problem by either 1) temporarily replacing ReLU with softplus in the first few epochs of training; or 2) warming up the weights by training a shallow (e.g., 5-layer) weight-tied stacking of $f_\theta$, then switching to MDEQ's equilibrium solver for the rest of the training.

Table 4: Settings & hyperparameters of each task. "cls." means classification task, and "seg." means segmentation task. These models coorespond to the ones reported in Tables 1, 2, and 3.

| | **CIFAR-10** (cls.) | | **ImageNet** (cls.) | | **Cityscapes** (seg.) | |
|---|---|---|---|---|---|---|
| | MDEQ-Small | MDEQ | MDEQ-Small | MDEQ-Large | MDEQ-Small | MDEQ-Large |
| Input Image Size | $32 \times 32$ | | $224 \times 224$ | | $1024 \times 512$ (train) $2048 \times 1024$ (test) | |
| Number of Epochs | 50 | 200 | 100 | 100 | 480 | 480 |
| Batch Size | 128 | 128 | 128 | 128 | 12 | 12 |
| Optimizer | Adam | Adam | SGD | SGD | SGD | SGD |
| (Start) Learning Rate | 0.001 | 0.001 | 0.05 | 0.05 | 0.01 | 0.01 |
| Nesterov Momentum | - | - | 0.9 | 0.9 | - | - |
| Weight Decay | 0 | 0 | 5e-5 | 1e-4 | 2e-4 | 3e-4 |
| Use Pre-trained Weights | - | - | - | - | Yes, from ImageNet | Yes, from ImageNet |
| Number of Scales | 3 | 4 | 4 | 4 | | |
| # of Channels for Each Scale | [8,16,32] | [28,56,112,224] | [32,64,128,256] | [80,160,320,640] | | |
| Width Expansion (in the residual block) | 5× | 5× | 5× | 5× | (Exact same model as in ImageNet) | |
| Normalization (# of groups) | GroupNorm(4) | GroupNorm(4) | GroupNorm(4) | GroupNorm(4) | | |
| Weight Normalization | ✓ | ✓ | ✓ | ✓ | | |
| # of Downsamplings Before Equilirbium Solver | 0 | 0 | 2 | 2 | | |
| Forward Quasi-Newton Threshold $T_f$ | 15 | 15 | 22 | 22 | 27 | 27 |
| Backward Quasi-Newton Threshold $T_b$ | 18 | 18 | 25 | 25 | 30 | 30 |
| Limited-Mem. Broyden's Method Storage Size $m$ | 12 | 12 | 18 | 18 | 18 | 18 |
| Variational Dropout Rate | 0.2 | 0.25 | 0.0 | 0.0 | 0.03 | 0.05 |

# B   Equilibrium Solving and Convergence Analysis

We extend our discussion on the convergence to equilibrium in Section 3.3 here. First, we briefly introduce the (limited-memory) Broyden's method that we use to perform the root-solving.

## B.1   (Limited-memory) Broyden's Method

As our goal is to solve the equation $g_\theta(\mathbf{z}^\star; \mathbf{x}) = f_\theta(\mathbf{z}^\star; \mathbf{x}) - \mathbf{z}^\star = 0$ for the (root) equilibrium point $\mathbf{z}^\star$ as efficiently as possible, an ideal choice would be Newton's method:

$$\mathbf{z}^{[i+1]} = \mathbf{z}^{[i]} - (J_{g_\theta}^{-1}\big|_{\mathbf{z}^{[i]}}) g_\theta(\mathbf{z}^{[i]}; \mathbf{x}); \quad \mathbf{z}^{[0]} = \mathbf{0} \tag{6}$$

However, in practice this involves two major difficulties. First, for a deep network with realistic size, the Jacobians are typically prohibitively large to compute and store. For instance, for a layer converting an input tensor of dimension $32 \times 32 \times 80$ (e.g., height × width × channels) to an output of the same shape, the resulting Jacobian will have dimension $81920 \times 81920$, which needs 25GB of memory to store. Second, even if we can store this Jacobian, inverting it would be an extremely expensive (cubic complexity) operation.

We therefore use a variant of Broyden's method [6, 5]:

$$\mathbf{z}^{[i+1]} = \mathbf{z}^{[i]} - \alpha \cdot B^{[i]} g_\theta(\mathbf{z}^{[i]}; \mathbf{x}); \quad \mathbf{z}^{[0]} = \mathbf{0} \tag{7}$$

where $\alpha$ is an adjustable step size and $B^{[i]}$ is a *low-rank approximation* to $J_{g_\theta}^{-1}\big|_{\mathbf{z}^{[i]}}$. Notably, we do not need to form the Broyden matrix $B^{[i]}$ explicitly, as we can write it as a sum of low-rank updates:

$$B^{[i+1]} = B^{[0]} + \sum_{k=1}^{i} \mathbf{u}^{[k]} \mathbf{v}^{[k]^\top} = B^{[0]} + UV^\top \tag{8}$$

where $\mathbf{u}, \mathbf{v}$ comes from the Sherman-Morrison formula [50]. We initialize the Broyden matrix to $B^{[0]} = -I$. As described in Section 3.2, we further extended Broyden's method with a limited-memory version that stores no more than $m$ low-rank updates $\mathbf{u}, \mathbf{v}$ each. Specifically, when the maximum storage memory $m$ is used, we free up memory by discarding the oldest update in $U$ and $V$ (other schemes are also possible).

## B.2   Discussions

**Runtime.**   The rate of convergence of MDEQ is directly related to the runtime of MDEQ. Because an MDEQ does not have "layers", a good indicator of computational complexity of MDEQ is the number of root-finding iterations (e.g., each Broyden iteration evalute $f_\theta$ exactly once). In practice, we stop the Broyden iterations at some threshold limit (e.g., 22 iterations), which usually does not yield the *exact equilibrium* (see Figure 6 and the discussion below). However, we find these estimates of the equilibria are usually good enough and sufficient for very competitive training of the MDEQ models. Similar observations have also been made in sequence-level DEQs [5].

(a) CIFAR-10 classification     (b) ImageNet classification     (c) Cityscapes segmentation

Figure 6: Plots of MDEQ's convergence to equilibrium (measured by $\frac{\|\mathbf{z}^{[i+1]}-\mathbf{z}^{[i]}\|}{\|\mathbf{z}^{[i]}\|}$) as a function of the number of times we evaluate $f_\theta$. As input image resolution grows (from CIFAR-10 to Cityscapes), MDEQ takes more steps to converge with (L-)Broyden's method. Standard deviation is calculated on 5 randomly selected batches from each dataset.

**Convergence on High-resolution Inputs.** As we scale MDEQ to higher-resolution inputs, the equilibrium solving process also becomes increasingly challenging. We identify at least two major reasons behind this phenomenon.

1. As the input resolution gets higher, so does the size of the Jacobian of $f_\theta$ which we try to approximate via Broyden's method. Therefore, more low-rank updates are expected for the Broyden matrix approximate the Jacobian and solve for the high-dimensional root.

2. Due to the nature of typical vision models, MDEQ employs convolutions with small receptive fields (e.g., the two $3 \times 3$ convolutions in $f_\theta$'s residual block) on very large inputs. To see how this complicates the equilibrium solving, consider a case where we simply iterate $f_\theta(\cdot; \mathbf{x})$ on $\mathbf{z}$ to reach the equilibrium point (i.e., not using Broyden's method; assuming $f_\theta$ is stable). Then we need *at least* as many iterations as required for the stacked $f_\theta$ to have a receptive field large enough to cover the entire image. Otherwise, new pixels covered by the larger receptive field will be available for each additional stack of $f_\theta$ (which disrupts the equilibrium).

This phenomenon is visualized in Figure 6, where we show equilibrium convergence of MDEQ models on CIFAR-10 (low resolution), ImageNet (medium resolution), and Cityscapes (high resolution) images by measuring the change of residual $\frac{\|\mathbf{z}^{[i+1]}-\mathbf{z}^{[i]}\|}{\|\mathbf{z}^{[i]}\|}$ with respect to calls to $f_\theta$. As with our experimental setting in Section 4, we initialize the Cityscapes MDEQ with the weights pretrained on ImageNet classification (pink line in Figure 6c). In particular, we observe that more Broyden iterations were required to reach the fixed point as the images get larger. For example, whereas MDEQ typically finds the equilibria with a good level of accuracy within 30 steps on CIFAR-10 images (cf. Figure 6a), over 100 steps are used on Cityscapes images (cf. Figure 6c).

Figure 7: Comparing MDEQ with single-stream DEQ on CIFAR-10. All resolutions of MDEQ converge *simultaneously* and in a much stabler way than the single-scale DEQ model. Larger scale index means higher resolution (e.g., "scale 1" is the highest scale).

Moreover, in all three cases, Broyden's method (blue lines in Figure 6) converges to the fixed point in a more stable and efficient manner than simply iterating $f_\theta$ (yellow lines), which often converges poorly or does not converge at all.

We find that the simultaneous multiscale fusion also effectively stabilizes the equilibrium convergence of an MDEQ. Figure 7 visualizes the convergence of all equilibrium streams (i.e., $\frac{\|\mathbf{z}_k^{[i+1]}-\mathbf{z}_k^{[i]}\|}{\|\mathbf{z}_k^{[i]}\|}$ for resolution $k$) in an MDEQ that is applied on CIFAR-10. For comparison, we also visualize the convergence of a single-stream DEQ [5] that maintains only the highest-resolution stream (i.e., $32 \times 32$). Specifically, from Figure 7 one can observe that: 1) all MDEQ resolution streams indeed converge to their equilibria in parallel; 2) lower-resolution streams converge faster than higher-

Figure 8: Examples of MDEQ-large segmentation results on the Cityscapes dataset.

resolution streams; and 3) high-resolution convergence is much faster in multiscale setting (pink line) than in the single-stream setting (orange line).

We hypothesize that Broyden's method and the multiscale fusion help with the equilibrium convergence because both techniques provide a faster way to expand the receptive field of $f_\theta$ (than simply stacking it). For Broyden's method (see Eq. (7)), the Broyden matrix $B^{[i]}$ is a full matrix that mixes all locations of the feature map (which is represented by $g_\theta(\mathbf{z}^{[i]}; x)$); whereas typical convolutional filters only mix the signals locally. On the other hand, multiscale up- and downsamplings broaden the effective receptive field on the high-resolution stream by direct interpolation from lower-resolution feature maps.

## C Qualitative Segmentation Results on Cityscapes

We demonstrate in Figure 8 some examples of the segmentation results of the MDEQ-large model (see Table 3) on Cityscapes (val) images (of resolution $2048 \times 1024$).