[Reviews · NeurIPS 2020]

Review 1

Summary and Contributions: This paper proposes a new implicit deep learning model, whereby the model does not have an explicitly deep architecture, rather it defines the 'neural network layers' implicitly as steps in internal solver (like Neural ODE). The work builds on top of existing work (Deep Equilibrium Models) by considering and solving for multiple scales jointly.

Strengths: - Great numbers in the experiments in realistic and complex data, considering that implicit models are elegant yet not competitive in such scenarios. Matching deep residual networks in datasets like ImageNet and Cityscapes is certainly an impressive feat. - Clear writing and exposition of the method. Reproducibility of the approach is probably quite possible.

Weaknesses: - After reading the paper, I honestly cannot really tell what is its 'core contribution'. Certainly, considering multiple scales is important, relevant, interesting. However, and unless I misunderstood, each resolution is treated independently, with its own loss and its own solver. (there is no interaction between any i and i+1 in equations 5, and in figure all the losses are independent to each other). While still important in that it brings a considerable improvement in accuracy, I would not call the proposed extended methodology a major technical contribution. - Continuing on the previous point, one of the important aspects of multiple scale learning is learning the interactions between multiple scales (eg, the eyes and the head are on different scales for person identification). If in the proposed approach each scale is treated independently and only in the end combined in a weak manner, to what extent is it possible the model to learn such interactions? - It is a bit unclear what contributes the most to the improvements in the results. There seems to be some differences as compared to the original DEQ models, and compared to Neural ODEs, in the way the preprocessing takes place. -- How important is the particular choice of downscaling the input, or of the linear projection layer prior to the input? -- Also, I might have missed it but which scales exactly are used in the various experiments? -- How is it made certain to keep the number of parameters equal? -- Equal splitting of the number of parameters oer scale among different scales? -- Which of all scales are the most important to deliver the final results? -- Is there a (very big) difference in accuracy with Neural ODEs just because of the downsampling/downscaling? - Some model choices are unclear. Why is x added only to the highest resolution? Is it a conscious design choice driven by a specific constraint, is it a practical choice in that it makes no difference when added to other scales, another reason?

Correctness: Yes.

Clarity: For the most part yes.

Relation to Prior Work: Good.

Reproducibility: Yes

Additional Feedback: Although there are unclarities, I will vote positively but I expect some clearly answers in the rebuttal.


Review 2

Summary and Contributions: As the main contribution, this work extends deep equilibrium models to the domain of images by devising a single-layer, multiscale architecture that learns multi-resolution image features. Unlike explicit models, that process different resolutions in succession, this method processes all feature scales simultaneously. Like its predecessor, this method is still able to represent an “infinitely” deep network, to perform inference based on a root solver, and to backpropagate considering only the layer at the equilibrium. As images are much larger than other domains previously exploited by implicit models, the authors also show how to optimize Broyden’s method to address high dimensionality issues. Finally, this is the first work to achieve competitive results on solid benchmarks using implicit models.

Strengths: - This work successfully builds upon deep equilibrium models and validates their use for real, large-scale datasets; - The method was compared to state-of-the-art implicit models and to well-established explicit models (e.g. ResNet, DenseNet); - The experimental evaluation was carried out in different datasets, including solid benchmarks (e.g. ImageNet, Cityscapes); - The authors demonstrate the weight transferability of their model, and its application for different problems(e.g. classification, semantic segmentation); - Experimental results were vastly superior to those obtained by other implicit models and on par with explicit models.

Weaknesses: - Considering that Table I and Figure 4(a) report results for CIFAR-10, it would be desirable to have results for the same group of explicit models in both of them. - The authors provided all hyperparameters in Table 4 (supplementary materials), but provide no intuition on how to set these values for different datasets (especially the parameters that are specific to the proposed method).

Correctness: Everything looks correct. Their experiments were very well designed and grasp different aspects of their method (e.g. accuracy, running time, different tasks, weights transfer).

Clarity: The paper is well written and well organized. Even though its content requires a broad background from the reader, it is easy to follow its insights, contribution, and empirical validation.

Relation to Prior Work: Yes, it is clear how this work builds upon Deep Equilibrium Models and how this work constitutes the new state-of-the-art for implicit models.

Reproducibility: Yes

Additional Feedback: Enjoyed reading the paper. Nice work,


Review 3

Summary and Contributions: This paper incorporates multi-scale processing into a new deep implicit model (the multi-scale deep equilibrium net, MDEQ) to rival the accuracy of the reigning deep explicit models for computer vision on large-scale tasks like ImageNet classification and Cityscapes segmentation. In contrast to deep explicit models, which define successive sequential stages of hierarchical features, the proposed MDEQ jointly solves for equilibrium across scales and the architecture consists of only one step of residual convolution and multi-resolution fusion by downsampling/upsampling all pairs of scales. With respect to implicit modeling, this work reconciles solving for the equilibrium with the practicalities of deep learning for vision like normalization, choice of nonlinearity, and receptive field size. To scale up to high dimensional image inputs, a custom solver based on Broyden's method is engineered. This work is the first proof that implicit modeling can be effective for vision, and surprisingly it brings into question the necessity of deeply hierarchical modeling as is currently ubiquitous in deep learning for vision.

Strengths: The approach is a significant departure from the norm for deep vision models. As surveyed in the related work, and represented in the explicit deep learning baselines, it is common for deep vision models to arrange many different depths and resolutions of features in sequence, each with their own parameters, to encode a hierarchical and multi-scale representation of images. When there is parallel processing, as on a pyramid, the interaction between scales is limited to simple aggregations like taking the sum or max. This work instead simultaneously solves for an equilibrium across multiple scales of a shallow transformation with only a single "stage" of parameters. This is an informative interrogation of the need for hierarchy, stages of distinct parameters, and depth vs. width in deep learning for vision. The method is novel with respect to existing implicit models. Prior methods do not incorporate multi-resolution fusion, or make implicit modeling compatible with normalization layers, dropout, and the effects of local operations like convolution with small filters. By exposing its multiple feature resolutions as an interface for optimization, the MDEQ can be used for multi-task learning or pre-training and fine-tuning in a way that NODE/ANODE and the DEQ cannot. The results are strong w.r.t. both implicit and explicit models. The single-stream and multi-scale DEQ both improve on the implicit baselines of NODE and ANODE on CIFAR-10, as examined in prior work. The large-scale results are a major step forward for implicit modeling on the scale of the input and the accuracy of the output: prior implicit models dealt with small images of 32x32 resolution (or less) while MDEQ goes up to 2048x1024, and the benchmark scores meet or approach those of the best and recent explicit deep networks. The results are relevant to research on implicit and explicit models alike, since the benchmark results are so strong. This work is the first bridging of the gap for practically-motivated, real-world vision.

Weaknesses: Implicit modeling still needs the right explicit architecture for the transformation that is driven to equilibrium. The MDEQ architecture is not entirely implicit, and stacks several transformations together. The input representation (l. 179-186) and the transformation's multi-resolution fusion step (l. 165-178) include several explicit layers, although the paper argues that it models just one layer (l. 223). - The input representation is computed by two convolutions, nonlinearities, and normalization that reduce the image input resolution by 4x. - The multi-resolution fusion stacks a convolution and nonlinearity for each 2x difference in resolution across features. - Furthermore, training is warmed up by first optimizing an explicit weight-tied architecture (five stacked replicas of the MDEQ) before switching to implicit modeling by root finding in a single MDEQ. It is impressive that implicit modeling with this "shallow" architecture is sufficient for strong results, but the explicit architecture is important, and that could be better acknowledged. There is a missing ablation of interest for architecture design: how effective is the tied-weight architecture, without the equilibrium solving, but instead unrolled in time for some number of steps? Such a stacked, tied-weight architecture is mentioned in passing as an initialization trick, but how accurate and efficient is it as a method in its own right? This result bears knowing on its own, because it could be that recursive convolution of this kind is under-appreciated, and knowing so would be an independent contribution of this work alongside implicit modeling. The computational costs are a trade-off that exchange less memory for more time. While this is a useful trade-off, since vision tasks can be memory-bound, it would be stronger still if solving for the equilibrium could be sped up. It must be noted that the proposed method is not so much slower, so this weakness is further lessened.

Correctness: All claims are substantiated and measured: accuracy on standard benchmarks, relative timing to a reasonable standard (ResNet execution time), and absolute memory usage. The pre-training and fine-tuning of the MDEQ is investigated for image classification to semantic segmentation transfer learning, as has been common in vision. The improvement over prior implicit models, in their existing experimental setting, and the competitiveness w.r.t. current explicit models, on common benchmarks, are both shown. The experiments and explanation are thorough and sound. - The baselines are recent and state-of-the-art for implicit modeling (ANODE) and explicit modeling (ResNets, DenseNets, high resolution nets, ...). - The benchmarks follow standard settings for image classification and semantic segmentation. - The supplement clearly identifies experiment details and confirms aspects of the method like convergence to equilibrium for single vs. multiple scales, the need for the optimizer vs. iterating the transform, and so forth.

Clarity: - The exposition includes clear text, figures, and notation. - The headings and grouping are sensible for comfortable navigation of the paper. - The paper neatly summarizes the deep equilibrium net on which it is based for encapsulation. - The code is included for inspection if there were points of confusion.

Relation to Prior Work: There is a thorough summary of implicit deep learning, both classic (recurrent backpropagation) and contemporary (neural ODE, ANODE, DEQ, OpNet, ...). The "challenges" paragraph (l. 135) underlines the extensions to the DEQ. There is good coverage of multi-scale modeling as done in vision, historically and at present, in both the introduction and background. While there are a variety of multi-scale architectures for vision, they either (1) process scales/resolutions in sequential stages or (2) process scales/resolutions in parallel but independently, as on a pyramid, with predictions a simple summary of the different scales. There are two points that could be better situated with respect to prior work: - It is certainly true that tied-weight architectures are not popular nor state-of-the-art in vision, but the claim "no such models exist in computer vision" needs qualification. There has been some exploration of weight-tied convolution [A] and reducing the number of parameters in explicit models by sharing parameters across a subset of the layers [B]. [C] recursively applies a convnet as a kind of recurrent model for semantic segmentation. Granted, there are no existing state-of-the-art models of this kind, and the degree of weight-tying in the proposed method is significantly greater. - The multi-resolution fusion step is new to implicit modeling, but not explicit modeling, as multi-scale architecture has been thoroughly explored for vision architectures. The related work does cover many architectures, but does not give specific credit to [54], which advanced the same all-scales fusion step for explicit deep models. This work is certainly a novel extension, in reducing the architecture to one weight-tied transformation of residual convolution and multi-resolution fusion, but more precise credit could be given. [A] Understanding deep architectures using a recursive convolutional network. Eigen et al. arXiv 2013. [B] ShaResNet: reducing residual network parameter number by sharing weights. Boulch. Pattern Recognition Letters 2018 [C] Recurrent Convolutional Neural Networks for Scene Labeling. Pinheiro and Collobert. ICML 2014.

Reproducibility: Yes

Additional Feedback: Final Review: Having read the author response and other reviews I am maintaining by positive rating. The results are a marked improvement on what has so far been accomplished by implicit nets for visual recognition. From the author response, it seems the unrolled tied-weight architecture is already strong, and solving for equilibrium further improves the results. I agree that this demonstrates the contribution, and urge the authors to include both results to encourage more research in both tied-weight architectures and implicit modeling. For Rebuttal: As an alternative to full implicit modeling, how effective would it be to unroll the model for a few steps and use it as a tied-weight explicit model? More specifically, the architecture could be stacked five times, and then the computation graph could be applied as usual for the forward and backward passes. The supplement mentions doing this as an initialization trick, but how well does this work at convergence? This would help characterize the use of the architecture apart from the use of solving for equilibrium. Either recursive convolution is helpful on its own, or the solver is required, but either is informative! In Figure 4 (a), even the DEQ (not MDEQ) improves on NODE + ANODE. Is this due to giving the DEQ + MDEQ the full resolution vs. smaller resolution? Is it computationally feasible to do at least one experiment with an ODE method at the full resolution for an even comparison? Additional Feedback: The summary of explicit deep models for vision as hierarchical and multi-stage is compelling and almost totally complete. Well done! The only exception worth further unpacking could be pyramid processing: with input pyramids, the modeling is parallel, but independent across scales, with only a simple summarization across scales such as taking the sum or max. The pyramid is a weaker form of parallelism than the MDEQ, as the pyramid is parallel and separate while the MDEQ is parallel and joint, but it is parallel. Re: multi-scaling modeling in convolutional nets (l. 101): there is a useful distinction to be made in explaining input pyramids vs. feature pyramids. Farabet et al. [20] apply their convnet in parallel across pyramid scales, but there is no interaction across scales in the network. I would not call this "built into convolutional networks" as much as a kind of pre-processing and post-processing. On the other hand [46] and especially [37, 54] combine features across scales in the network as a kind of feature pyramid.


Review 4

Summary and Contributions: This paper proposes a multi-scale equilibrium model MDEQ. Similar to previous implicit models, MDEQ uses implicit differentiation to reduce the memory usage during training. However, MDEQ also leverages multiple image resolutions to improve the performance. In addition of conventional CIFAR experiments, it also show very promising results on large-scale datasets (ImageNet/Cityscape).

Strengths: 1. The results are quite impressive: although it is still behind state-of-the-art explicit models, it outperforms previous implicit models by a quite large margin (Table 1). 2. The idea of using multi-resolution is intuitive.

Weaknesses: I am not an export in "implicit" models, but I wonder: 1. Why do you need to keep the all resolutions throughout, causing the slow runtime. If you reduce the resolution, would it achieve both high accuracy and low latency? 2. In addition of training latency/memory, how about inference latency for MDEQ?

Correctness: The paper mainly claims the contribution of "multi-scale features" in implicit models. Although "multi-resolution" has been widely used in explicit models, it seems to be not common (and non-trivial) in implicit models, so I don't have any concerns about the claims. The methodology is intuitive and reasonable, and verified by the strong results.

Clarity: Yes, the paper is well written.

Relation to Prior Work: N/A

Reproducibility: Yes

Additional Feedback: The core idea of this paper is about multi-resolution, however it is unclear why and how multi-resolution is helpful. It would be better to do some ablation studies, such as: 1. Comparing w/o and w/ multi-resolution using the same model and setting; 2. Comparing different types of multi-resolutions. Assuming the original image is 224, you can compare {224, 112}, {224, 112, 56}, {224, 112, 56, 28}, etc. In this way, we can better understand how multi-resolution helps the final accuracy and memory cost.

[Author Response · NeurIPS 2020]

We thank all reviewers for their valuable and positive feedback, and share their excitement that MDEQ validates implicit
models for large-scale vision applications. We address a few comments below, which we will incorporate into the paper.

**Review #1.** We first want to clarify a misunderstanding here: each resolution's representation is **not** treated indepen-
dently. While they have their own residual blocks (which Eq. (5) describes), the features of each resolution are all
subsequently mixed in the multi-resolution fusion step (Fig. 1). Both components (residual block and multiscale fusion)
are in the $f_\theta$ that MDEQ iterates. Hence importantly, instead of having "independent losses and solvers", the whole
MDEQ system has *only one solver*, and the equilibria are solved *simultaneously* (see L187-204 and Appendix C), with
loss induced on resolution $i$ can flow to the parameters of resolution $j \neq i$ via implicit differentiation. In other words,
the multi-resolution interaction itself is a part of the transformation driven to the equilibrium. Indeed, to answer the
reviewer's question, this is one of the primary contributions of MDEQ, and we hope this clarifies the confusion.

As a result, MDEQ is different from the original DEQ models and the Neural ODE models (which are both single-
stream (see Fig. 3)). However, we found the particular choice of downsampling not important. For instance, we didn't
downsample CIFAR-10 images at all, and downsampled for Cityscapes only to small degree by 2 convolutions (see Table
4 for "# of Downsamplings before Equilibrium Solver"). In contrast, NODEs typically downsample MNIST/CIFAR
images by $4\times$ before ODE solvers, because the high-dim ODEs can be challenging to solve. The scales we used and
the splitting of parameters can both be found in Appendix A (under "Hyperparameters" and Table 4); the parameters
across models were kept equal by adjusting the "Width Expansion" factor in the residual block.

Finally, there is no difference in performance (empirically) for injecting the input x to 1) the highest resolution *only*; or
2) to *all resolutions*. In fact, we started off with the latter design but then simplified the model to become the #1 case
that we use now. Besides this minimal empirical influence, injecting only to the highest resolution both *reduces* the
need for extra parameters (i.e., one needs to downsample the signals to lower resolutions first, before injecting them to
these resolutions) and *simplifies* the model by removing redundant (explicit) pre-processing of the input.

**Review #2.** The original purpose of Figure 4(a) was only to compare implicit models of the same size ($\sim$170K
parameters, which was used by most ODE-based models but rarely by explicit networks like ResNet-18). But we are
happy to add the curves for larger MDEQ and larger explicit models to Figure 4(a) as suggested by the reviewer. In
terms of the hyperparameters shown in Table 4, we mostly followed the hyperparameters used by prior works such as
HRNet [54] (e.g., weight decay, optimizer, etc.), while the threshold of Broyden iterations were adjusted according to
the size of the original input – larger images typically need more quasi-Newton iterations to converge (see Figure 5).

**Review #3.** We absolutely agree and acknowledge that in practice, rather than "a layer", what MDEQ used to iterate
in the implicit solver is actually more of a *block*, and will definitely clarify this language. As the reviewer identified,
we found that at the finest level of MDEQ, such mini explicit transformation is still very helpful in terms of the
representation driven to equilibrium. Regarding the downsampling of the original raw image, as mentioned above, its
primary utility is to reduce the runtime of the model, rather than increase the accuracy. However, we do agree with R3
that the usage of explicit structure is still helpful in both *the design* and *the training* (e.g., shallow warmup) of MDEQ,
and we will clarify this further in the paper.

Per the suggestion of the reviewer, we also performed an additional ablation study using a weight-tied (unrolled) version
of an MDEQ(-large) layer on ImageNet (we used 5 layers, which is the max our GPU can fit). This model eventually
achieved 75.8% accuracy, as compared to MDEQ-large's 77.5% (which we believe validates the benefit of modeling an
"infinite layer"). Regarding the original DEQ's improvement on NODE/ANODE, we generally found that the usage
of initial downsampling not critical for the final results on CIFAR-10. Empirically, even with the same $4\times$ smaller
resolution, we are still able to achieve $87\%$ accuracy with MDEQ-small and $93.5\%$ with MDEQ, which is the same
level of performance as currently reported in Table 1. When using full-resolution, we found that ODE-based models are
significantly slower (especially the original NODE), and full resolution ANODE still reached only 60.3%.

**Review #4.** The rationale behind keeping all resolutions throughout were: 1) Having access to all resolutions is the
key factor enabling MDEQ to be pretrained (e.g., on ImageNet) and subsequently finetuned (e.g., on Cityscapes) on
very different resolution "interfaces", a common setting for many pattern recognition problems. 2) Having multiple
scales side-by-side is actually one of the ways for us to reduce latency (rather than increase it), which we discuss
in depth in Appendix C (and in Figure 6). Driving multiple streams to the equilibrium stabilizes the convergence to
equilibrium, making the Broyden convergences typically faster.

To further investigate the effect of multi-resolution (i.e., multiple stream) itself, we have also compared the MDEQ we
used with single-stream DEQ models in the paper. As can be seen in Table 1 and 2, the single-stream DEQ (which is
essentially MDEQ with 1 resolution) performs substantially worse than when modeling multiple resolutions (though
still much better that NODE/ANODE models). Figure 6 also shows that the convergence to the equilibrium is also
much slower at the highest resolution. We can run further ablations investigating the effect of the number of resolutions.

[Meta-Review · NeurIPS 2020]

This paper proposes a multi-scale equilibrium model that shows that implicit models can potentially compete with explicit models. Their work built on top of a previous proof-of-concept and solve many technical difficulties to achieve competitive performance on challenging image datasets. The paper is clear and the method is reproducible.